# The values and meanings of social activities for older urban men after retirement

**Risa Takashima**[1], **Ryuta Onishi**[2], **Kazuko Saeki**[3,4], **Michiyo Hirano**[2]*

**1** Department of Rehabilitation Science, Faculty of Health Sciences, Hokkaido University, Sapporo, Hokkaido, Japan, **2** Department of Comprehensive Development Nursing, Faculty of Health Sciences, Hokkaido University, Sapporo, Hokkaido, Japan, **3** Faculty of Nursing, Toyama Prefectural University, Toyama, Toyama, Japan, **4** Hokkaido University, Sapporo, Hokkaido, Japan

* mihirano@med.hokudai.ac.jp

**Data Availability Statement:** Our interview data contains potentially identifying and sensitive participants' information. Therefore, there is an ethical limit to sharing the data publicly. Anonymized excerpts from the full transcripts can

## Abstract

Previous studies have indicated that older men often experience disconnection from the community after retirement. Social activities have been shown to be effective in preventing social isolation among older urban men. Nevertheless, it has been reported that they often do not participate in community social activities and tend to be reluctant to do so. We explored the values and meanings of social activities for retired older men living in an urban area of Japan to understand support using social activities that are more suitable for them. Semi-structured interviews were conducted with 15 older men (aged 68–80 years; M = 74.6 ± 3.79 years) about their interactions with family and non-family members, and their participation in various community social activities. The grounded theory approach was used for the analysis. As a result, the following five categories were derived as the values that participants place on the social activities that they engage in: "health as a resource and reward for social activities," "feeling I am still useful," "feeling that something is my responsibility," "feeling of time well spent," and "finding interest through interactions." In addition, the following three categories were extracted as meanings of social activities: "fulfilling social life," "maintaining stable family relationships," and "maintaining safety and peace in the community." When considering the social activities that older urban retired men are interested in and likely to participate in, these five values can be considered indicators. In contrast, to maintain stable family relationships and safety and peace in the community, participants sometimes used strategies to stop or abandon social activities. Therefore, in situations where a peaceful life within a family or neighborhood is threatened, it may be useful to help set aside sufficient time and allow for psychological leeway in advance to incorporate social activities into their lives.

## Introduction

Participation in social activities improves the health of all older adults from multiple perspectives and ultimately results in lower mortality [1–3] and higher survival rates [4]. Several observational studies suggest that engaging in social activities provides the following benefits:

be made available to qualified researchers by request to the ethical committee of the Faculty of Health Sciences, Hokkaido University, who can be contacted at shomu@hs.hokudai.ac.jp.

**Funding:** This work was supported by JSPS KAKENHI [grant number JP18H03103]. In addition, we received support from Nihon Unisys, Ltd. for research expenses. The former had no role in study design, data collection and analysis, decision to publish, or preparation of the manuscript. A researcher from Nihon Unisys, Ltd. helped collect interview data.

**Competing interests:** This study received unrestricted research grants from Nihon Unisys, Ltd. A researcher of Nihon Unisys, Ltd. played a role in data collection. This relationship does not alter our adherence to PLOS ONE policies on sharing data and materials.

increased happiness [5]; good physical and/or cognitive functioning [5–8]; half the risk of admission [2]; and reduced risk of disability in activities of daily living, instrumental activities of daily living, and mobility [9, 10]. In contrast, an increased risk of disability has been reported in those who do not engage in social activities [11].

For older men, because the central focus of their life would have changed from their workplaces to the community after retirement, participation in social activities could be meaningful for them. In particular, the present cohort of Japanese older men comprise the generation that has contributed to Japan's economic growth from the post-war period to the present, through long working hours under the permanent employment system [12]. A majority of their adult working lives were spent in their workplaces; after retirement, participation in social activities in the community may help them adjust to additional free time.

Recently, the Japanese government has been promoting social participation as one of the precautionary approaches for older people [13]. Community-based centers (i.e. "salons") using social activities have been recommended as a strategy to promote the health of older people [14]. These "salons" are being implemented as places where older people can participate in the community and also engage in various social activities. For example, older people living near these salons gather to create arts and crafts, play games, and interact with preschool children [14]. Continuous participation in "salons" can prevent disability [15] and cognitive decline [14]. However, men's participation in salons is significantly lower than that of women [15, 16]. This is not unique to salons—the frequency of older men's participation in social activities within the community is lower than that of women [17, 18]. Despite the benefits of participating in social activities, older men often do not and do not want to participate in social programs in the community [19].

In particular, retired and unemployed men are at a higher risk of social isolation and loneliness than are employed men [20, 21], and retirement may contribute to the social isolation experienced by older men. Loss of work may lead to the loss of colleagues and networks, social support, and autonomy [22]. Retired older men are also less involved in meaningful activities in the community [23, 24] and are often disconnected from the community [23]. Especially in urban areas, older men have been reported to be more isolated than older women [18, 25]. Social isolation predicts functional decline among men in urban areas, even if they go out every day [26]. In contrast, social connections with friends and neighbors protect the functional health of older men [27]. Participation in social activities can be effective in preventing social isolation among older urban men [28].

To help older urban men after retirement engage in valuable social activities in their communities, it is necessary to understand the values and meanings of social activities for them. Therefore, we asked the following research question to explore support in the community using social activities more suitable for older men: What are the values of social activities, and what are their meanings for retired older men living in an urban area of Japan?

## Methods

### Definitions of terms

The word "value" is defined as "relative worth, utility, or importance" [29], while the word "meaning" is defined as "significant quality; especially: implication of a hidden or special significance" [30]. In this study, the "value" of social activities refers to the importance or usefulness attributed to social activities by participants. The "meaning" of social activities refers to the quality and significance of the social activities, engaging in which makes the participants' lives seem to be important and have purpose. The difference between these terms, in this context, is that "value" is an attribute of a social activity, whereas "meaning" relates to a participant's sense of purpose pertaining to a social activity and the importance they place on it.

Social activities can be classified into informal social activities (i.e. with family, friends, relatives, and local residents) and formal social activities (e.g. participation in volunteer organizations), depending on the person(s) involved [31]. In previous studies incorporating this conceptual classification, informal social activities were found to protect functional health [27], suppress depression [32], and contribute to life satisfaction [33], as compared to formal activities. Formal social activities suppress cognitive decline as compared to informal activities [34]. In addition, Levasseur, Richard, Gauvin, and Raymond [35] categorized social activities into six levels according to their level of involvement with others and society. The term "social activity" in this study includes formal and informal work, and refers to the following activities at levels 2–6, excluding the preparatory stage in the study by Levasseur and colleagues: being with others (alone but with people around), interacting with others (social contact) without doing a specific activity with them, doing an activity with others (collaborating to reach the same goal), helping others, and contributing to society.

## Design and context

This study utilized the grounded theory approach—a qualitative methodology aimed at building a theory rooted in data that allows researchers to be flexible and allow additional data as needed to saturate the resulting categories [36]. This study, as well as the grounded theory approach of Corbin and Strauss, was based on the theoretical premises of interactionism and pragmatism, which are greatly influenced by John Dewey and George Mead [36].

According to Blumer [37], interactionism refers to a form of interaction that occurs between people. This theory holds that humans are not just organisms that respond, but living organisms that act, and that not only interpret responses to factors that interact with them, but also create actions based on what they consider. In other words, it emphasizes that actions are not viewed as being evoked inside the actor by outside, but as being created by the actor. In addition, according to Dewey [38], knowledge comes from acting or interacting with self-reflective beings. Meaning is captured, generated, and exploited through experience and never brought in from outside sources beyond reality [38]. Furthermore, there is no dualism conflict for pragmatists, and it is impossible to separate emotions from actions; they flow together with one leading into the other [38].

This study aimed to understand the value and meaning that older men ascribe to their social activities, how they engage in social activities, and the phenomenon from the viewpoint of those who act. Interactionism and pragmatism were used as theoretical premises to provide a perspective for understanding, interpreting, and redefining the empirical world of the actors in analyzing their narratives from their point of view.

## Participants and recruitment

This study was performed in one ward in a Japanese metropolis. The population of this district is about 290,000. Purposeful sampling was adopted for the recruitment. Inclusion criteria for study participants were men aged 65 years or older, after retirement or semiretirement, living in the ward, with the ability to speak and experience engaging in any social activity.

We approached the members of a cooking class for local older men, which was conducted in this ward. This cooking class is planned and held by a neighborhood association about three times a year, and about 50 middle-aged and older men participate each time. About 10% of the participants were aged younger than 65 years. Three researchers participated in this cooking class, explained the purpose of the research, and recruited participants. Thirteen older men agreed to participate.

As discussed above, participation in social activities is beneficial in preventing functional impairment. In contrast, analysis suggests that functional impairments may alter the

experience of participating in social activities. Therefore, to improve study rigor and trustworthiness, we included older men from the same community who needed assistance as negative cases. We explained the purpose of the study and the inclusion criteria to the organizers at the Community Comprehensive Care Centre in this ward and asked them to introduce us to older men in need of assistance. Two older men who needed assistance agreed to cooperate. Typical sample sizes for grounded theory research range from 10 to 60 persons [39]; 15 older men participated in this study. The characteristics of 13 participants recruited from cooking classes are shown in Table 1. Note that all the participants' names are pseudonyms.

## Data collection

The first author individually interviewed eight participants, the second author interviewed four, and the last author interviewed three between October and December 2018. The first author is an occupational therapist, and the second and last authors are public health nurses. The first and last authors, both women, hold PhDs, while the second author (a man) holds an MSc. All the interviewers are university researchers, who have practical experience in the field of healthcare related to older people and experience conducting qualitative research.

Participants initially reported their name, age, family structure, and life history including occupational history on the demographic questionnaire. Interviews were then conducted in the privacy of their homes or in one of the public community facilities, depending on the

Table 1. Participants' demographic profile.

| Participants' pseudonym | Age (years) | Years after retirement | Family composition | Role in neighborhood associations | Social activities that participants described (*Refer to the left column regarding neighborhood associations) |
|---|---|---|---|---|---|
| Ito | 75 | 10 | Only couple | Participant | Helping farmers, patrolling schools, park golf, drinking parties with his friends, fishing with friends, *Hyakunin Isshu* (One Hundred Poems by One Hundred Poets) at a children's hall |
| Watanabe | 76 | 14 | Only couple | Director | Citizen college for older people, walking events at former workplace |
| Yamamoto | 80 | 20 | Only couple | Participant | Checking around the parking lot, table tennis at a senior citizens' club, park golf |
| Nakamura | 79 | 8 | Only couple | Director | Golf, drinking parties with friends, karaoke, light sports |
| Kobayashi | 79 | 16 | Only couple | Director | Watching children on their way to and from school, senior citizens' club, amateur radio, interaction with former colleagues, gathering with high school classmates |
| Kato | 69 | 1 | Only couple | Director | Watching children on their way to and from school, paid and free volunteers, lunch with friends |
| Yoshida | 74 | 14 | Couple and their children | Director | Activities of welfare commissioners, old boys' associations of former workplace, drinking parties with friends |
| Yamada | 72 | Semi-retirement | Only couple | Director | Activities of welfare commissioners, karaoke with friends, class reunion |
| Sasaki | 73 | 10 | Only couple | Participant | Activities of old boys' associations, Advisory activities at corporate research institutes, reading in libraries, sports in gymnasiums, drinking parties with friends, social exchange at a favorite Japanese pub |
| Yamaguchi | 69 | 8 | Only couple | Director | Interaction through blogs, sign language circles, coffee schools, organizing events planned by oneself |
| Matsumoto | 68 | 3 | Only couple | Director | Activities as a committee member at schools and nursing care facilities, cooking class and karaoke club in the neighborhood, sports in gymnasiums |
| Inoue | 78 | 8 | Couple and their grandchildren | Participant | Management of *shigin* class, having meals with students of the *shigin* class, sports in gymnasiums, hot spring with friends |
| Kimura | 80 | 12 | Couple and their children | Director | Senior citizens' club |

participant's request. For four participants, their wives were home; therefore, they were present for part or all of the interview, depending on the participant's request.

The interviews revolved around their daily, weekly, and monthly lives; interaction with their families; interaction with non-family members; and participation in various social activities in the community. Participants were asked to reflect on their life in the past year. They were also asked about their interactions with other people and their participation in activities. Following this, they were asked about their perceived societal connection including and with others outside of their families. The interview guide is shown in Table 2; however, the interview guide was used as a flexible tool, and participants' free speech was respected. All the interviews were conducted in Japanese (the researchers' and participants' native language), recorded, and transcribed verbatim. Although the transcripts were not returned to the participants, the results, including relevant narrative descriptions, were shared with them, and they provided feedback. Excerpts from the interviews that appear in this paper were translated to English from the original Japanese transcripts.

This study followed the guidelines set out by the 1975 Helsinki Declaration (2008 revision). It was approved by the Ethics Review Committee of Hokkaido University Faculty of Health Sciences (no. 18–39). Participants were given verbal and written explanations on the assurance of anonymity, the confidentiality of data, the assurance of free participation in research, and the publication of the results. All participants provided written consent for research cooperation. They were free to withdraw at any time. After the interviews, participants were given souvenirs and sweets to show gratitude.

## Data analysis

The first author read the transcribed interview data carefully, removed the data using natural cuts in the manuscript as cut-off points, and coded each data. The participants talked about various activities, but the narratives corresponding to level 2–6 social activities defined by Levasseur, Richard, Gauvin, and Raymond [35] were coded. The coding did not involve simple paraphrasing—it involved elevating raw data to a conceptual level [36]. By identifying the properties and dimensions of the data, the characteristics and variations of the data were obtained and coded. According to them, properties define and describe a concept, and dimensions refer to the diversity within a property. Based on properties and dimensions, coded concepts were compared, and related concepts were categorized into higher-level concepts. Three levels of categorization were conducted after coding. Following this, the relationships between top-level categories were explored using a diagram or a visual device that depicts the

**Table 2. Interview guide.**

| Questions (translated to English from the original Japanese interview guide) |
| --- |
| 1. First, could you think about your daily life for a day, a week, or a month? |
| 1) Please tell me about your interactions with your family members in detail. |
| 2) Do you interact with people other than your family members? Please tell me about it in detail. |
| 3) Do you ever join a social gathering or organization in the community? Please tell me about it in detail. |
| 2. Second, could you think about your life for a year? |
| 1) Please tell me about your interactions with your family members in detail. |
| 2) Do you interact with people other than your family members? Please tell me about it in detail. |
| 3) Do you ever join a social gathering or organization in the community? Please tell me about it in detail. |
| 3. What do you think about connecting with society? |
| 4. What do you think about connecting with people other than your family members? |
| 5. Is there anything you are working on for your wellbeing? |

relationships between analytical concepts [36]. In all of these analytical processes, researchers repeatedly used questioning, making comparisons, and memo writing as analytical strategies [36]. All data analysis was conducted manually.

The first author initially categorized the interview data for five participants and modified it until consent was obtained from the three other collaborators. Next, the data for 10 participants were categorized based on the concepts drawn from the data of the five participants and revised again until consent was obtained from the three other collaborators. Two of the 10 patients who needed assistance were negative cases and were compared with the narratives of the other participants. Then, data from each participant was added until it was completed for all 15 participants. Variations were discovered starting from the twelfth participant's data, but nothing new was added at the level of the conceptualized category. This led the researchers to conclude that theoretical saturation had been reached.

## Study rigor and trustworthiness

The following strategies were used to increase the trustworthiness and credibility of data analysis [40, 41]. Several times, the researchers joined a cooking class attended by the participants and spent a certain amount of time with them. They performed continuous comparative analysis and described the results of the analysis process in detail. Community-dwelling older men, including the research participants, attended a social gathering for preventive care where the findings of this study were shared. A member check was also conducted in a community gathering where participants took part in. The researchers explained the results of the analysis, including the participants' key narratives, and received feedback from the participants. One interesting comment was that, according to participants, the results were accurate in describing their vague feelings. In addition, negative cases [36] were added to the analysis and compared with other participants.

## Findings

Semi-structured interviews were conducted with the aim of clarifying the research question: What are the values of social activities, and what are their meanings for retired older men living in an urban area of Japan? Fifteen older men, including two people in need of support as negative cases, participated. They were aged 68 to 80 years ($M = 74.6$, $SD = 3.79$). One was semi-retired, and 14 had been retired for 1 to 20 years ($M = 9.1$, $SD = 5.65$). As for family structure, one man was living alone, 11 men lived with their spouse, and three men lived with their spouse and children or grandchildren. Nine men were members of the neighborhood association, which were described as representative social activities. Other social activities mentioned are shown in Table 1.

As a result of the analysis, 541 codes were extracted, and the following five categories were derived from the codes as the values that participants place on the social activities that they engage in: "health as a resource and reward for social activities," "feeling I am still useful," "feeling that something is my responsibility," "feeling of time well spent," and "finding interest through interactions." In addition, the following three categories were extracted as meanings of social activities: "fulfilling social life," "maintaining stable family relationships," and "maintaining safety and peace in the community." Table 3 shows the list of features of categories, categories, and subcategories.

### The five values of social activities

Participants place the following five values on social activities: "health as a resource and reward for social activities," "feeling I am still useful," "feeling that something is my responsibility,"

**Table 3. Category structure.**

| Features of categories | Categories | Sub-categories |
| --- | --- | --- |
| Values of social activities | Health as a resource and reward for social activities | Health as a resource for important social activities |
| | | Involuntary reduction of social activities due to poor health |
| | | Strategic use for physical and psychological health maintenance |
| | Feeling I am still useful | Demonstrating my knowledge and abilities |
| | | Sense of being useful to people and society |
| | Feeling that something is my responsibility | Sense of mission that I must do |
| | | My turn |
| | | Awareness of "selflessness" contributions to society |
| | Feeling of time well spent | Worthwhile challenge |
| | | Social activities that cannot begin or last unless the core of my interest is shared |
| | | Pursuit of a favorite |
| | | Revenge for my past grievances |
| | Finding interest through interactions | Creating exchange opportunities that make my life interesting |
| | | Joy of interacting with friends who can share my way of life |
| | | Fun of socializing with friends that is comfortable for me |
| | | Pleasantness of an equal relationship without a title |
| Meanings of social activities | Fulfilling social life | Daily sense of fulfillment |
| | | Realizing importance of interaction through losing it |
| | | Needs for social activities created by the voids in life |
| | Maintaining stable family relationships | Maintenance of stable family relationships |
| | | Selection of social activities according to the busyness of family roles |
| | Maintaining safety and peace in the community | Conscious association for a peaceful life |
| | | Building connections in the neighborhood in case of emergency |

"feeling of time well spent," and "finding interest through interactions." These values were important criteria in determining whether participants engaged in social activities. These five categories will be described one by one.

First, as stated in "health as a resource and reward for social activities," health is both capital for participating in social activity and a value obtained from participating in social activity. In the other four categories, there was a relationship where participants engaged in social activities because of the value represented by those categories. In contrast, this category included value as a resource for engaging in social activities as well as value as a possible reward for engaging in social activities. Yoshida said, "*the neighborhood association has no age restrictions; so, you can take part in it while you are healthy.*" Ito said that he must be "*healthy*" to continue helping on the farm. In contrast, participants also talked about experiences of unwillingly abandoning or reducing social activities due to poor health. Tanaka, a negative case, was repeatedly hospitalized and had to turn over his paint shop to his son. "*I feel like I'm being told to resign, that I'm being given an indou by my disease,*" said Tanaka, still unable to forget his work. *Indou* is a term that comes from Buddhist traditions, and literally means that the person receiving the *indou* should "put an end to something." In modern times, however, the term is

mostly used metaphorically. He said that unlike the time when his health was at its peak, he could no longer go with his younger brothers and friends to pick vegetables.

At present, many participants have used social activities strategically to maintain physical and psychological health. "*Going there. . . well, you know. . . I still go even when I'm tired,*" said Yamamoto, who attends a club of older people twice a week. "*I still go. In spite of. . . I don't know how to say this; but I feel like I don't get sick.*" Yamaguchi shared about falling into depression after retirement and how he used social activities to defeat it.

> "*I was trying to get out (of the condition). I was trying to somehow get out of my depression. In the words of my wife, there was discord inside of me, and it was so subtle that even my wife didn't notice it. I didn't express it. I can laugh talking about it now; but that conflict really made me feel sad. I didn't have the will to do anything. (Omitted) I've always wanted to join the neighborhood association. I thought it would be one way to defeat my depression.*" (Yamaguchi)

Rising above his depression through social activities, Yamaguchi now updates his blog every day and enjoys planning and managing events to disseminate his knowledge of movies.

Second, participants valued the ability to "feel that they are still useful" through social activities. Yamada, who continues to monitor older people as a welfare commissioner, shared why he continues to participate in social activities. A welfare commissioner is commissioned by the Minister of Health, Labour and Welfare in each community to respond to inquiries from the perspective of the citizens, provide necessary support, and strive to improve social welfare. These people work for free as volunteers [42].

> "*When I started doing it, I was gradually waking up (to reality). A lot of people go through this process. I've had people approach and thank me. Some people thank me for helping them some time ago. These words are really refreshing. I keep doing my best, and whenever people thank me, I gradually feel refreshed and energized.*" (Yamada)

Participants also valued the use of their knowledge and abilities. Ito, who lends a hand to farmers, shared his feelings about instances where his advice helps farmers.

> "*It's a completely different profession, isn't it? But these are farmers who have used the same antiquated methods. I give out advice on how to improve practices. I'm able to give advices from a different perspective. They've adopted a lot of it. It's a lot of work, not only the methods, but also in all sorts of tasks. At the end of the day, I think it's a positive influence for both sides. (Omitted) I think I've helped them plenty.*" (Ito)

Third, if someone had to do it, participants engaged in social activities with a "feeling that something is my responsibility." Kobayashi, who is involved in fire prevention activities, said that no one wanted to take on the job; so, he had to rise to the occasion. Yoshida explained how he started volunteering in the community: "*There aren't many people who want to volunteer. I happened to be close to the chairman of the neighborhood association; so, I couldn't refuse.*" In contrast, Yoshida realized that "*in two or three years, I would need support, so I thought about helping while I still can.*" While he was conscious that someone would take over the role someday, at that time, he was engaged in social activities because it was his duty to support it.

Fourth, participants valued the "feeling of time well spent." Expressing his feelings towards volunteering, Kato said, "Everyone *was doing their best; so, I felt like I should do my best too.*"

Yamamoto, who cracks down on illegal parking in the apartment building where he lives, emphasized the seriousness of his activity saying, "*I'm really doing my best*." Watanabe, who organizes the neighborhood association's events, compares his experience as a guest and organizer, and describes the pleasure of planning and organizing events by themselves.

> "*(Before) I joined the events, well, as a guest. Now, we are the ones planning the events. It's a different experience. Well, once you put in the work. . . well, it's. . . it's hard; but it's fun.*" *(Watanabe)*

When asked about the recreational activities he enjoys, Watanabe said, "*I know that I should find and engage in something, anything. (Omitted) Whenever I go to public facilities, I bring a lot of pamphlets with me.*" However, he stated, "*It's more and more often becoming a hassle (to do it).*" He said he was eager to work on challenging social activities, while he was looking for recreational activities to fill the void in his life but felt bothered.

Lastly, "finding interest through interactions" was also a characteristic value that older men place on social activities. When asked about why he continues to interact with members who he met at local sports competitions, Nakamura said, "*Members could become advisers about something.*" He adds that he can exchange information on local hospitals with other people, saying, "*There are things that other people teach me and things that I teach them.*" Matsumoto, a care councilor, explained the value of this social activity:

> "*Even though I am a care council member, I also learn new opinions and experiences from people for whom I provide care. It's like reading a book. You talk to people, listen to their opinions, and a different person springs out of them. I think it's the same as reading a book.*" *(Matsumoto)*

Matsumoto talked about finding interest in interacting with people using this metaphor. Regarding the values that participants found in socializing, the comfort of socializing without the need for labels was noted. Yoshida expounds this in the following:

> "*It feels like we're back to square one where your past job doesn't matter. Some of us were school principals, doctors, and police officers. There's also a university professor; but he comes more irregularly now because he is old. All these people gather, and everyone is on equal footing. No titles. It's what makes socializing interesting. . . (Omitted). . . Some people try to flaunt their titles; but the group immediately shuts them out.*" *(Yoshida)*

Doing away with these titles means that the participants themselves are not bound by these words. It also influences people's manners when they interact in the community. Participants found the new ways of interacting in communities outside the professional hierarchy interesting.

### The three meanings of social activities

The participants were engaged in social activities for "fulfilling social life." After retirement, a void appeared in the social aspects of the participants' lives that their work had previously occupied. If a participant's life is likened to a "box," it can be imagined that a void was created in the box. This life box had quantitative and qualitative aspects. The quantitative aspect represented time, and the qualitative aspect represented social significance meaning to remaining a social being. For example, sharing about the temporal void in his life, Kobayashi, who enjoys interacting with people through an amateur radio, said, "*Actually, I did have free time*," when

asked about why he does it. Watanabe who had battled depression for several years after retirement expounded on the qualitative void in his life: "*You should try to believe that you're living a meaningful life. Otherwise, gradually, you'll be left alone. This could be boring, which could make you feel depressed.*" Through engaging in social activities, participants experienced a "fulfilling social life"—temporally and significantly speaking—which resulted in the "filling of their social-life box."

Some participants were conscious of the size of their boxes and adjusted so that the void would be eliminated. Kobayashi stated that he had planned to start a neighborhood association activity before and after his retirement: "*Our neighborhood association isn't strict. We welcome everyone who comes here. (Omitted). . . I've always thought about helping. . . about starting a neighborhood association after quitting my job.*" Matsumoto shared how social activities came in as his work faded out of his lifestyle:

"*I retired at 60. After that, I worked as a contract employee for five years before quitting. Work was easier as a contract employee; so, I thought it would be a good idea to start a neighborhood association. That was six years ago. I started the association with the vice chairman, and for a period, I became the chairman.*" (Matsumoto)

In "maintaining stable family relationships," some participants talked about using social activities to maintain a moderate distance between themselves and family members and forging good family relationships. Family members living in remote areas rarely affected the social activities of participants. In contrast, cohabiting family members often influenced participants' engagement in social activities. Sasaki, who used to teach in a university, explained that he would go out to distance himself from his wife:

"*In short, wives are often irritated when their husbands retire and spend more time at home. That's why, even if that's not the case for us, I stay at home in the morning and sometimes go out in the afternoon.*" (Sasaki)

Sasaki tried to go to the library or the gym every day to make sure that he and his wife can spend their time alone.

In contrast, during family issues, since family relationships were top priority, some participants abandoned social activities for "maintaining stable family relationships." Suzuki, who was a negative case and needed to take care of his wife who had lost most of her sight, said he let go of low-priority social activities because, "*I have already done them.*" If he becomes single, Suzuki said he wants to play Go or ballroom dance again. However, he explained, "*If I don't have a wife, my mind may change again; but. now, I feel like my life revolves around my wife.*" With his wife at the center of his world, there is simply no room for new social activities in Suzuki's social-life box.

In some cases, participants engaged in social activities for "maintaining safety and peace in the community." Without the connections Nakamura has made during peacetime, he would have said, "*They aren't my business. It's okay if nobody comes to check on them.*" Matsumoto explained that smooth cooperation during emergencies requires people to forge connections during peacetime to prepare for such situations:

"*To provide immediate response in a disaster, you must first know people. If I don't know what my neighbor is doing, even if I said I could go there to lend them a hand, I don't think I would be much of a help. That's why these connections—human connections—are very important.*" (Matsumoto)

Social activities were used to create these connections during peacetime. In contrast, participants said that while connections are important to maintain a peaceful life, it is also important to avoid crossing certain boundaries. They stated that they were concerned about building a good neighborhood without crossing boundaries.

> "*If the people in the community were connected, I think it would be nice to be able to freely talk, greet and personally meet when something comes up. I think that kind of socializing is good; but I don't want a very close relationship with them.*" (Ito)

> "*When you talk about unnecessary things with your neighbors, there's a chance that the story may change. (Omitted)... That's why I rarely talk that much about something.*" (Yamamoto)

In addition, social activities were sometimes temporarily abandoned for "maintaining safety and peace in the community" as well as "maintaining stable family relationships." In this study, participants were interviewed from October to December 2018; however, in September 2018, there was a large earthquake in the area where they lived, and participants experienced a power outage for two to three days. When the natural peace of life in the area was threatened, participants said that they focused on restoring peace and not on social activities.

## Discussion

This study explored the values and meanings of social activities for retired older men who live in an urban area of Japan. Five and three categories were derived as the values and meanings that participants place on the social activities that they engage in, respectively. To understand their values and meanings of social activities for these older men, it is necessary to consider the impact of retirement on their lives. Retirement creates a significant void in the lives of participants in the dimensions of time and social significance. In addition, the center of participants' lives will shift from the workplace to the community. Engagement in social activities appears to play an important role in helping older male participants adjust to life in the community. Moreover, this process of adjustment may be interpreted as a hypothetical concept—"filling my social-life box"—that might link the 8 categories.

To help understand the concept of "filling my social-life box," a conceptual diagram illustrating the structure of quantitative/qualitative fulfilment in retirement life using social activity among retired older men was created (Fig 1).

The concept of "filling my social-life box" might relate the eight aforementioned categories to each other. If the participants' lives were likened to a "box," it seems that two conditions were found to lead the participants' to engage in social activities. The first condition was that the social activity had one of the following five values: "health as a resource and reward for social activities," "feeling I am still useful," feeling that something is my responsibility," "feeling of time well spent," and "finding interest through activities." The second condition was that the participants had a temporal void in their life box or felt that it might occur in the future. As described in the categories of "maintaining stable family relations" and "maintaining safety and peace in the community," the participants' life boxes did not have room for social activities in situations where they had other activities to prioritize. To maintain control over their lives, participants used strategies to stop or abandon social activities. As consequences of their engagement in social activities, participants reached "fulfilling social life," "maintaining stable family relations," and "maintaining safety and peace in the community." However, as mentioned above, the latter two categories also seemed to serve as prerequisites for engaging in social activities. In addition, the participants were likely to increasingly realize the value of

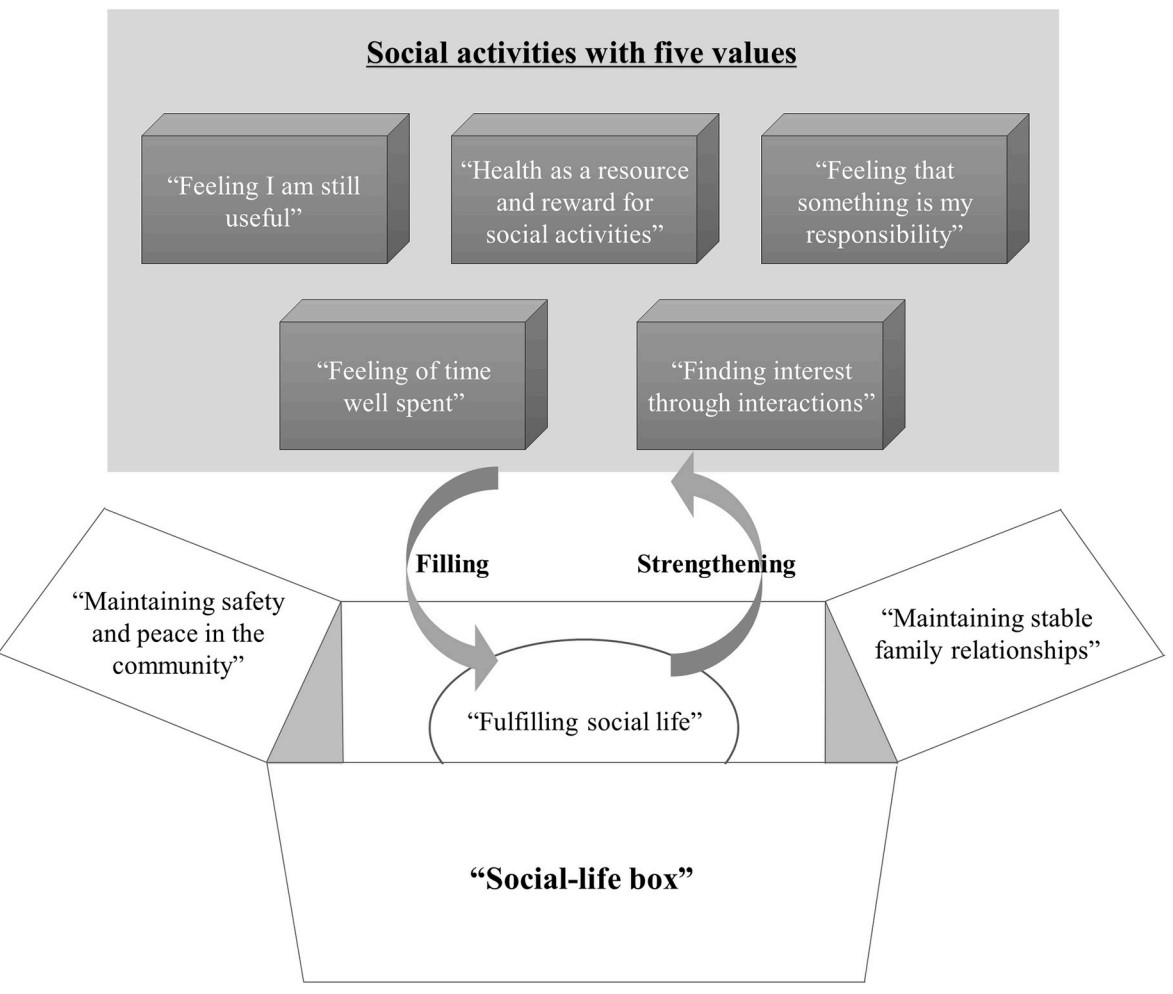

**Fig 1. Filling one's social-life box using social activities in older urban men after retirement.**

these activities by engaging in valuable social activities; consequently, they might strengthen their engagement in these activities and pursue engagement in further activities.

## Characteristics of the meanings in social activities for older urban men

Throughout life, people have different time resources and demands, which are constantly changing as they get older [43]. Retirement is a transition in life that ensues forcible changes in people's use of time [44]. For many, work is the center of their lives, and retiring from work and restructuring of life after retirement are important issues in old age. Of the 15 participants in this study, 14 had retired (an average of 9.1 years after retirement) and 1 was semi-retired with reduced workload. For the participants, engaging in social activities was a way to fill quantitative time and qualitative social significance.

Retirement is a process rather than an event [45]. Genoe, Liechty, and Marston [46] assessed the retirement experience of older men in Canada and found that there were three stages to the transition to retirement: pre-retirement, characterized by both apprehension about retirement and idealization of the perfect retirement; the initial transition, which participants compared to an extended vacation, but in which they also struggled to adjust to increased amounts of free time; and mid-transition, when participants learned to balance

structure and flexibility. In our study, quantitatively and qualitatively fulfilling participants' lives with social activities may have helped participants adapt to their retirement community life. It can be understood that filling one's social-life box with social activities is a process of adapting to the increase in free time in the "initial transition stage."

The "mid transition stage" requires balancing activities in their lives. A situation in which valuable activities in one's life are balanced is described as the individual's perception of having the right amount of activities and the right variation between activities [47]. In other words, too few or too many activities may lead to an imbalance in life. When comparing men and women, the time allotted to housework for men is annually increasing and becoming more equal to that of women [48]. Nevertheless, men still spend less time doing housework than do women [49]. In other words, the social-life box of men is more likely to have more quantitative voids than women. Some people are fortunate enough to pick what activities they engage in; but this may also be a challenge. In fact, time spent on housework is positively correlated with health among both men and women [49]. Social activity may have been a useful tool for men in addressing the quantitative void that appears after retirement.

In addition, "filling my social-life box" may have provided participants with a real sense of meaning in life (MiL). Bartrés-Faz et. al. [50] describes MiL based on the following three components: purpose in life, sense of coherence, and engagement with life. Among them, engagement with life refers to the individual's evaluation of how important, worthwhile, and inherently valuable their life feels and the sense of having a life worth living. Bartrés-Faz et al. [50] further notes that this component covers the relevance of perceived significance and fulfillment of one's life and the importance of engaging in valued and meaningful activities. Participants in this study achieved "filling my social-life box" through engaging in social activities valued from five perspectives. This could be similar to achieving engagement with life through engaging in valued and measurable activities. Furthermore, of the five values of social activities, "feeling I am still useful" and "feeling that something is my responsibility" may be related to the realization of purpose in life, which is a component of the MiL. Purpose in life refers to having a strong sense of meaning and future-oriented goals in life [50]. The participants' sense of purpose in life might be strengthened by engaging in social activities where they could feel as if they are still useful and that something is their responsibility.

Maintaining a peaceful life within the family and in the neighborhood was the most important and basic aspect of participants' post-retirement lives. Social activities were utilized for these. In contrast, social activities were sometimes abandoned to maintain and recover them. This is because activities aiming to maintain and improve safety and peace occupy a large amount of the social-life box. Some participants quit or stopped engaging in social activities; thus, participants prioritized safety and peace over social activities, which is consistent with the values of older Japanese people. In a survey of 1,870 men and women aged 60 and older living in Japan, 93.1% said they would stay in their current community [49]. In addition, when asked about what is needed to sustain a peaceful life, most responses pointed to "mutual support between neighbors" (55.9%), followed by "support from family and relatives" (49.9%) [49]. As a prerequisite to shift one's attention to social activities, it is important that the foundations such as family and community are stable as a place for social activities for older men in the community [20].

In situations where peaceful living in the family or in the community is threatened, closing the lid of one's social-life box and avoiding social activities is a strategy that participants employed to help maintain the right quantitative balance. This means that, if the social-life box is quantitatively full of high-priority or urgent activities, there is no room for social activities, even if the void is not filled in terms of social significance. For example, nursing care has been reported to limit social activity [51, 52]. Without the deferment in the quantitative time

dimension, any problem related to the significance dimension may be inevitably postponed. Choosing what, when, how, and how much activity to take on in response to a changing situation will reflect the adjustment in maintaining a balance between valuable activities [53]. The participants' interview responses indicated that they were trying to fill their social-life box after retirement and balance higher priority activities.

Comparing the social activities of men and women in terms of significance in a time use survey among 60- to 74-year-old French people (N = 10,764), it was reported that women were more active in nursing care and informal social exchanges, while men were more active in volunteering and community activities [54]. The results of this study indicated that there might be a social significance void in the post-retirement lives of the older men. Considering the social significance aspects of the space created in their social-life box after retirement, it can be assumed that it typically included elements such as a feeling of time well spent, social contribution, and a sense of responsibility. These factors are consistent with three of the five values that participants held in this study—"feeling I am still useful," "feeling of responsibility," and "feeling of time well spent." Consequently, it can be said that social activities were those that effectively filled the portion of the social-life box that was occupied by work in terms of significance.

However, this is not necessarily exclusive to men. A survey regarding the time of high-achieving women after retirement shows that they spend time volunteering in community service organizations as well as nursing and spare-time activities, and some participants have reported volunteering as a primary retirement activity [55]. This suggests that the social activities that older people want to perform depend not only on gender, but also on one's previous lifestyle.

The category "maintain safety and peace in the community" could be redefined at the community level as a concept of social capital. To date, several classification systems for social capital have been suggested, of which the concept of bonding and bridging social capital has credence [56]. According to Szreter and Woolcock [57], bonding social capital refers to aspects of "inward-looking" social networks that reinforce exclusive identities and homogeneous groups (e.g., age, ethnicity, and social class). In contrast, bridging social capital refers to "outward-looking" social networks across different social and ethnic groups that do not necessarily share similar identities. For "maintain safety and peace in the community," through their various social activities, participants appeared to be trying to build forms of bonding and bridge social capital. In particular, participants emphasized that they should avoid crossing certain boundaries when connecting with neighbors. This might be a strategy for the participants to build bridging social capital smoothly.

In addition, Saito et al. [58] identified the following three subscales for community-level social capital: civic participation, social cohesion, and reciprocity. The social activity for "maintain safety and peace in the community" carried out by the participants of this study could be significant to foster social cohesion, which is a factor strongly related to trust, community trust, and attachment [58]. Nine of the participants in this study were directors of neighborhood associations, which may have been influenced by their high interest in improving social cohesion. Noguchi et al. [59] reported that living in communities with high social cohesion was associated with a decreased incidence of functional impairment in older men. Moreover, Amemiya et al. [60] showed that in communities with high social cohesion, older men who perceived that their communities' social cohesion was high showed greater functional ability improvement than men who perceived it to be low. In their study, functional ability was assessed through levels of disability, which were objectively assessed at the time of certification for the utilization of long-term care (LTC) services, based on nationally standardized criteria in Japan. There are seven levels of disability: Requiring Support-1 and -2 and Requiring LTC-l

(partial support needed for basic activities of daily living) to LTC-5 (complete support needed for all activities of daily living). The social activities for "maintain safety and peace in the community" may not only benefit the community by increasing the social cohesion but may also benefit the functional health of older men living in the community, including the participants themselves.

## Characteristics of the values that older urban men associate with social activities

In this study, participants chose social activities based on the following values: "health as a resource and reward for social activities," "feeling I am still useful," "feeling of responsibility," "feeling of time well spent," and "finding interest through interactions." One of the pioneering social activities in the community for men is Men's Sheds, which is becoming popular internationally [61]. Men's Sheds have demonstrated a certain effect as a place for social activities for older men in the community [20]. Men's Sheds are usually community spaces where older men can interact while participating in various woodworking and other activities [62]. Men's Sheds provide an unstructured environment for male fellowship and valuable activities [63]. In other words, Men's Sheds have a positive impact on the health and well-being of older men by providing a space where valuable activities take place [22]. As men tend to be less involved in community activities than women, Men's Sheds are a place where older men are more likely to participate in social activity in communities.

While the concept of Men's Sheds tends to focus on the social inclusion of older men, Wilson and colleagues [64] found that 37% of Men's Sheds were also actively promoting health. In the results of this study, health was a key factor influencing social activities, which supported the results of previous studies. In contrast, participants barely mentioned social inclusion in this study—they were more concerned about social withdrawal as this could have a negative effect on their health. Most participants had already participated in social activities in the community, and this may be because social inclusion was not a chief concern.

Lefkowich and Richardson [65] revealed the experience of older men participating in Men's Sheds in Ireland. As a result of their interviews, four key characteristics were revealed: using and developing new skills, feeling a sense of belonging, supporting and being supported by peers, and contributing to community. We consider the use and development of new skills and their contribution to the community to be similar concepts to "feeling I am still useful" and "feeling that something is my responsibility." In contrast, the participants did not talk about the sense of belonging, providing support to peers, and receiving support from peers. It is presumed that members of Men's Sheds share activities and time with the same members, which strengthens the bond between members and makes it easier to feel a sense of belonging. In this study, 13 of the 15 participants participated in the same cooking class for men. However, we believe this difference occurred because there were several variations in the social activities that the participants mentioned in the interviews.

Watanabe [66] revealed that retired older men noticed that they had no connection to the community and had participated in exchange activities because they wanted to interact with people in other communities. Our participants also valued the connection with people and local communities, but as expressed in "finding interest through interactions." It is noteworthy that social interactions did not seem to require emotional connections. Older people living in rural areas are often idealized as people having strong ties to their communities and having maintained high quality relationships with their friends for decades [67]. In contrast, older people in urban areas may not be required to have strong ties with their communities. As stated by participants in "maintaining safety and peace in the community," it is important to

protect peaceful living, especially in the community. However, the lack of emphasis on emotional connections between peers in social activities might suggest the impact of the urban environment in this study.

Participants in this study sought to find something interesting through interacting with people rather than forge emotional connections. In a qualitative interview survey on successful ageing of men and women (59% women) over the age of 60 years living in urban areas of the United States, social interactions were a key concept that constituted the main theme—"life/self-growth" [68]. Participants mentioned that social interaction is a learning process, and that it develops when information from experienced people is incorporated [68]. Despite not using the term "life/self-growth," the current participants were interested in interacting with people who had various experiences as a source of learning. This was consistent with the results of the previous study. Additionally, participants in this study found that it was interesting to interact in communities where professional titles did not matter, unlike in companies. They also mentioned that those who flaunt their social status were immediately disliked in the community. This type of interaction meant that participants could be relieved from thinking about their social status. Concurrently, this became the usual conduct when interacting within the community. This feeling may be characteristic of retirees who have experienced situations that put a premium on social status. One of the most important factors in the social activities of older men is experiencing something interesting through interactions that lead to personal growth and interactions with new relationships that they did not experience in the corporate world.

## Limitations and further research

The United States and Canada have gradually abolished mandatory retirement since the 1970s, and the United Kingdom and European Union countries have abolished mandatory retirement since the 2000s [69]. In contrast, in Japan, salaried workers, who make up the majority of workers in urban areas, generally retire at the age of 60–65 [70]. In particular, the participants of this study were from the generation that had worked under the permanent employment system; that is, for their entire lives, they worked at the same organizations [12]. In pre-retirement life, it is likely that not only formal, but also informal social life is completed in their workplace settings. The participants of this study engaged in social activities for "filling my social-life box," which might have been strongly influenced by their context, which perhaps required them to build a new social life in the community after retirement. The results of this study may have transferability to other older adults whose social lives mostly revolve around their workplaces.

Older men are less likely than women to participate in community social activities [17, 18], and older men are reported to be less likely to participate in community social programs [19]. In contrast, the participants in this study were recruited from community cooking classes. Therefore, the results of this study should be understood as the experience of older men who are interested in and have participated in community social activities. However, two participants who were negative cases and recruited outside of community cooking classes tended to have less social activity than the other 13 participants. One of the two said that he had put social activities on the back burner because of "maintaining stable family relations," while the other said that he had to give up his participation in social activities because of functional impairments. Because this study aimed to explore the meaning and value of social activities, the focus was on the experiences of older men who were engaging in social activities. Further research should explore the experience of older men who do not actively engage in social activities.

Older male rural farmers were reported to have used two strategies to adapt to the retirement process: "gradual transition-tapering off" and "maintaining connections" [71]. In

contrast, for the participants in the current study, engaging in social activities was a strategy to adapt to the retirement process. In other words, while Wiseman and Whiteford's study [71] described that retirement for farmers occurs gradually, jobs such as urban office workers generally have a compulsory retirement. These differences in the manner of retirement may have affected participants' experiences with social activities. In addition, previous studies on the relationship between living in urban or rural communities and social activity are inconsistent. However, there are also reports stating that rural older adults have less social activities than do their urban counterparts [72, 73]. Further research is needed on the differences in social experience brought by retirement styles as well as areas of residence.

When examining social activities that are comfortable for older urban men, it may be useful to consider health, contribution, responsibility, worthwhileness, and interest as mentioned in this study. "Maintaining stable family relationships" and "maintaining safety and peace in the community" were sometimes given priority over "fulfilling social life." Therefore, when providing support using social activities, consideration should be given to the prerequisite that the family and the community are at peace. Based on these findings, we plan to develop a social activity program for community-dwelling older men and conduct empirical research.

## Conclusion

This study explored the values and meanings of social activities for retired older men who live in an urban area of Japan. The following five categories were derived as the values that participants place on the social activities that they engage in: "health as a resource and reward for social activities," "feeling I am still useful," "feeling that something is my responsibility," "feeling of time well spent," and "finding interest through interactions." When considering the social activities that older urban retired men are interested in and likely to participate in, these five values can be taken as indicators. In addition, the following three categories were extracted as meanings of social activities: "fulfilling social life," "maintaining stable family relationships," and "maintaining safety and peace in the community." As described in the categories of "maintaining stable family relations" and "maintaining safety and peace in the community," the participants' lives did not have leeway for engaging in social activities in situations where they had other activities to prioritize. To maintain control over their lives, participants used strategies to stop or abandon social activities. Therefore, in situations where a peaceful life in a family or neighborhood is threatened, it may be useful to help set aside sufficient time and allow for psychological leeway ahead of time to incorporate social activities into their lives.

## Acknowledgments

We offer special thanks to the participants who cooperated with this study. In addition, we thank Mr. Joao Carlos Koch Junior of Sapporo Gakuin University, Hokkaido, Japan for content editing and proofreading this manuscript.

## Author Contributions

**Conceptualization:** Risa Takashima, Ryuta Onishi, Kazuko Saeki, Michiyo Hirano.

**Data curation:** Risa Takashima.

**Formal analysis:** Risa Takashima.

**Funding acquisition:** Michiyo Hirano.

**Investigation:** Risa Takashima, Ryuta Onishi, Michiyo Hirano.

**Methodology:** Risa Takashima, Ryuta Onishi, Michiyo Hirano.

**Project administration:** Michiyo Hirano.

**Resources:** Risa Takashima.

**Supervision:** Kazuko Saeki, Michiyo Hirano.

**Validation:** Risa Takashima, Ryuta Onishi, Kazuko Saeki, Michiyo Hirano.

**Visualization:** Risa Takashima.

**Writing – original draft:** Risa Takashima.

**Writing – review & editing:** Risa Takashima, Ryuta Onishi, Kazuko Saeki, Michiyo Hirano.

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
