## [Decision Letter · Decision Letter 0]

1 Sep 2020

PONE-D-20-21202

The values and meanings of social activities for older urban men after retirement

PLOS ONE

Dear Dr. Hirano,

Thank you for submitting your manuscript to PLOS ONE. After careful consideration, we feel that it has merit but does not fully meet PLOS ONE’s publication criteria as it currently stands. Therefore, we invite you to submit a revised version of the manuscript that addresses the points raised during the review process.

We look forward to receiving your revised manuscript.

Kind regards,

Sze Yan Liu, PhD

Academic Editor

PLOS ONE

Journal Requirements:

"This work was supported by JSPS KAKENHI [grant number JP18H03103]. In addition,

we received support from Nihon Unisys, Ltd. for research expenses. The former

sponsorad had no role in study design, data collection and analysis, decision to

publish, or preparation of the manuscript. A researcher from Nihon Unisys, Ltd. helped

collect interview data."

We note that you received funding from a commercial source: "Nihon Unisys, Ltd"

Reviewers' comments:

Reviewer's Responses to Questions

**Comments to the Author**

1. Is the manuscript technically sound, and do the data support the conclusions?

Reviewer #1: Partly

Reviewer #2: Partly

2. Has the statistical analysis been performed appropriately and rigorously? 

Reviewer #1: N/A

Reviewer #2: Yes

3. Have the authors made all data underlying the findings in their manuscript fully available?

Reviewer #1: No

Reviewer #2: Yes

4. Is the manuscript presented in an intelligible fashion and written in standard English?

Reviewer #1: Yes

Reviewer #2: Yes

5. Review Comments to the Author

Reviewer #1: This is an interesting article focusing on the values and meanings of social activities for older urban men after retirement. Using a qualitative approach, the authors logically illustrated the inside of older men. The story is very clear, and the manuscript is well written. To enhance the quality of this study, I would like the authors to revise the following points.

1. Page 1, the authors’ name. The last letter of the authors’ family name may be unnecessary. For example, Takashimaa should be replaced to Takashima.

2. Page 4, paragraph 3. I do not understand why you needed to show the information regarding Men’s Sheds here. To me, this specific information does not fit in the Introduction. You could add this kind of information as an example in the Discussion where you mentioned this topic.

3. Page 6, line 93. The term “examine” seems to be too strong in the qualitative study. Usually, this term is used in the quantitative study. It would be better if you select an alternative verb.

4. Pages 6-7, definitions of terms. If you talk about the definitions in this study, this information should be in the Methods section.

5. Page 7, lines 112-118. You showed the levels of social activities. Did you evaluate the participants using these levels? If yes, how did they affect the extracted categories? You could show this information in the table 1. If no, I do not know why the authors introduced this information here.

6. Page 8, participants and recruitment. Why did you choose the cooking class among various social activities? Is there any possibility that older men in that cooking class are very different from ordinary older men? What do you think about their representativeness? You could, at least, mention this point as a limitation of this study.

7. Pages 8-9, lines 138-144. You added two community-dwelling older adults who needed assistance. Were they treated as negative cases in this study? If yes, I think that you do not need to include them in table 1. Or, they were treated as the participants under the different criteria? If you intended to analyze older adults with various conditions, it would be better to change the methodology of recruitment.

8. Pages 9-11, table 1. You have the information regarding years after retirement and family composition. Did they affect the participants’ social participation? If yes, how?

9. Page 15, line 213. How was a member check conducted?

10. Page 16, line 245. What do you mean by “using” in the category “using and obtaining health through social activity”? Can using health and obtaining health be in the same category? The name of this category seemed to be a different nature compared with other categories.

11. Pages 24-25, storyline. The authors mentioned two components in the social-life box: temporal void and significance void. I do not know if they come from the study results. Which category belongs to which void? I understand that one of the aims of a qualitative study is developing a hypothesis. However, this social-life box story seems to be a logical leap to me. In my opinion, this story did not explain the detailed interactions or hierarchy among categories sufficiently. For example, was there any participant who were reluctant to participate in the cooking class at the beginning? For older men, a cue of social participation maybe not always active. It would be more interesting if you show the complex interactions among categories.

12. Fig 1. The sharpness is not enough.

Reviewer #2: Thank you to the authors for an interesting paper on older men’s experiences and perceptions on the value and meanings of social activities. I enjoyed reading the paper and learning about older Japanese men's views on social activities. There are, however, some areas that need to be addressed in order for this to make a strong contribution to the literature:

1. The authors have indicated they have used Grounded theory as the methodology, but they also need to explain the epistemology (e.g. constructionism) and theoretical perspective (e.g. symbolic interactionism) that framed their study. This will help to justify their choice of grounded theory to address their research question, and situate their research more broadly.

2. Please clarify if the data analysis conducted manually or via a program such as nVivo.

3. It would also be helpful to provide a table that documents how one or more of the categories were derived. This would help to more clearly distinguish some of the categories. In particular, the ‘values’ category of ‘feeling they are still useful’ has a strong overlap with the ‘meanings’ category of ‘fulfilling social life’ – particularly in relation to ‘social significance’. When looking at these different categories, the distinction between values and meanings seems somewhat artificial, and tends to obscure the real relevance of the data.

4. The identification of the core category of ‘filling a social-life box’ provides an interesting and novel way of interpreting the data, but I’m not convinced that it adequately reflects the value and meaning of social activities for the participants, nor that it accurately conveys how social activities are implicated in older men’s health and well being.

5. My main concern is that in striving to derive theory through data, the authors have overlooked a significant body of literature that is very relevant to the issues raised in their study. As such, they have not adequately situated their theory of ‘filling a social-life box’ in the literature – this is a key requirement of grounded theory studies. Moreover, I get a strong sense that the analysis has ‘reinvented the wheel’ – that is, the authors have developed a theory on the values and meanings of social activities among older retired men that aligns closely with relevant literature and theory, but have not adequately acknowledged that literature, or attempted to situation their findings within that literature. In particular, the authors should consider how their findings relate to literature on social capital and social support among older men. For example, their discussion on how participants engaged in social activities to ‘maintain safety and peace in the community’ relates closely to the concept of social capital, and the notion that community relationships can be leveraged as a ‘commodity’ to draw on in times of stress and need. Indeed, Matsumoto’s comment about why human connections are important reflective on the value of social capital. Significantly, social capital works at both the individual and community level, and can act to support the health of men who are not socially connected, as well as those who do have strong social connections. Other aspects of social capital that are relevant to the findings of this study include the notion of bonding and bridging capital – this relates to Yamamoto’s comment on community connection. Therefore, this paper would be considerably strengthened by integrating a discussion on how their findings link to the literature on social capital in the Discussion section.

6. Likewise, the categories identified in this study such as ‘feeling that something is my responsibility’ and ‘feeling they are still useful’ have direct relevance to the literature on meaning and purpose among older people. Again, this paper would be considerably strengthened by integrating a discussion on how their findings link to existing literature in this area.

7. There are some problems with English grammar – please carefully proof read the paper.

6. PLOS authors have the option to publish the peer review history of their article (what does this mean?). If published, this will include your full peer review and any attached files.

Reviewer #1: No

Reviewer #2: No

---

## [Author Response · Author response to Decision Letter 0]

17 Sep 2020

I have uploaded the letter that responded to each point raised by the academic editor and reviewers as a separate file labeled 'Response to Reviewers', according to the instructions in the decision letter.

---

## [Decision Letter · Decision Letter 1]

21 Oct 2020

PONE-D-20-21202R1

The values and meanings of social activities for older urban men after retirement

PLOS ONE

Dear Dr. Hirano,

Thank you for submitting your manuscript to PLOS ONE. After careful consideration, we feel that it has merit but does not fully meet PLOS ONE’s publication criteria as it currently stands. Therefore, we invite you to submit a revised version of the manuscript that addresses the points raised during the review process.

I agree with both of the reviewers' assessment that your revised manuscript has greatly improved.  However, there are several minor points that still need to be addressed.

We look forward to receiving your revised manuscript.

Kind regards,

Sze Yan Liu, PhD

Academic Editor

PLOS ONE

Reviewers' comments:

Reviewer's Responses to Questions

**Comments to the Author**

1. If the authors have adequately addressed your comments raised in a previous round of review and you feel that this manuscript is now acceptable for publication, you may indicate that here to bypass the “Comments to the Author” section, enter your conflict of interest statement in the “Confidential to Editor” section, and submit your "Accept" recommendation.

Reviewer #1: (No Response)

Reviewer #2: (No Response)

2. Is the manuscript technically sound, and do the data support the conclusions?

Reviewer #1: Partly

Reviewer #2: Yes

3. Has the statistical analysis been performed appropriately and rigorously? 

Reviewer #1: N/A

Reviewer #2: Yes

4. Have the authors made all data underlying the findings in their manuscript fully available?

Reviewer #1: No

Reviewer #2: No

5. Is the manuscript presented in an intelligible fashion and written in standard English?

Reviewer #1: Yes

Reviewer #2: Yes

6. Review Comments to the Author

Reviewer #1: Thank you for revising the manuscript based on my comments. To enhance the quality of your article, I would like to suggest to you for some minor revisions as follows.

1. Page 1, the authors’ name. The last letter of the authors’ family name may be unnecessary. Onishib could be Onishi, Saekic could be Saeki, and Hiranob could be Hirano. Please ensure the accuracy of each expression very carefully.

2. Page 2, line 25. The verb “examine” sounds too strong in the qualitative study. You could change this to another verb if you agree.

3. This time, you emphasized “filling my social-life box” as the core category. However, this change brought me a feeling of strangeness. “Filling my social-life box” seems to be a hypothetical concept rather than a category extracted from the data. I understand the importance of this concept in your study line. However, your main focus should be to explore the values and meanings of social activities as you mentioned in the introduction. I recommend you to treat “filling my social-life box” just as an idea of your interpretation. Otherwise, readers would have an impression of a logical leap.

Reviewer #2: Thank you to the authors for their comprehensive response to the reviewer comments. I am satisfied with their response to my comments.

There are just a few outstanding issues that need attending to:

1. Methods, Design and context section

a. I don’t understand what the authors mean by the phrase – “Furthermore, there is no binomial (??) conflict for pragmatists, and it is impossible to remove emotions from actions, and actions have an emotional response” (p. 8 line 125-127) – could the authors please simplify this statement to clarify their meaning?

b. Likewise, the first sentence in the next paragraph also needs rephrasing to enhance meaning – “This study aimed to understand the value and meaning of older men in social activities”. Do the authors mean “This study aimed to understand the value and meaning that older men ascribe to their social activities”? (p. 8, line 128)

2. Discussion, Characteristics of the meanings… section

a. p. 30, lines 487- 488 - the sentence reads “…may be related to the realization of purpose in life, which is the component of the MiL”. I suggest this be changed to - “…may be related to the realization of purpose in life, which is a component of the MiL”.

b. P. 33, lines 541-543 – I don’t think the participants said that they sought to foster both bonding and bridging social capital! This needs to be rephrased to indicate that through their various social activities, participants appeared to be trying to build forms of bonding and bridging social capital.

c. P. 33, line 548 – should one of Saito et al., subscales be ‘recipient’? This doesn’t seem to align with the other subscales that are listed here (civic participation and social coesion).

d. P. 33, lines 554-557 – sentence beginning “Moreover, Amemiya … showed greater functional ability improvement than men who perceived it to be low” – to help the reader understand this, it would be useful to provide an example of functional improvement.

e. Pp 33-34 – last line on p. 33 and first line on p. 34 – sentence beginning “The social activities for…” – I’m not sure what the authors mean by “could have both self- and community-interest for the participants”. Could the authors please clarify what they mean by self and community interest, in the context of the broader discussion in this para on social capital?

7. PLOS authors have the option to publish the peer review history of their article (what does this mean?). If published, this will include your full peer review and any attached files.

Reviewer #1: No

Reviewer #2: No

---

## [Author Response · Author response to Decision Letter 1]

4 Nov 2020

Following the editor's instructions, I submitted another file labeled 'Response to Reviewers'.

---

## [Editor Report · Decision Letter 2]

11 Nov 2020

The values and meanings of social activities for older urban men after retirement

PONE-D-20-21202R2

Dear Dr. Hirano,

We’re pleased to inform you that your manuscript has been judged scientifically suitable for publication and will be formally accepted for publication once it meets all outstanding technical requirements.

Kind regards,

Sze Yan Liu, PhD

Academic Editor

PLOS ONE

Additional Editor Comments (optional):

Thank you for the correcting the minor grammatical revisions.  In addition, thank you for taking the reviewers' suggestions regarding the vocabulary used in specific sentences.  Your manuscript is more clear as a result. 
---

## [Editor Report · Acceptance letter]

13 Nov 2020

PONE-D-20-21202R2 

The values and meanings of social activities for older urban men after retirement 

Dear Dr. Hirano:

I'm pleased to inform you that your manuscript has been deemed suitable for publication in PLOS ONE. Congratulations! Your manuscript is now with our production department. 

Kind regards, 

on behalf of

Dr. Sze Yan Liu 

Academic Editor

PLOS ONE